# ROS-mediated apoptosis in colon cancer cells induced by sulfated polysaccharides from *Gracilaria corticata*

Zeynab Falihzadeh[1], Laleh Roomiani[2]*

1 MS.C Department of Food Science and Technology, Ahv. C., Islamic Azad University, Ahvaz, Iran,
2 Department of Fisheries, Ahv. C., Islamic Azad University, Ahvaz, Iran

* Laleh.roomiani@iau.ac.ir

## Abstract

### Background

Sulfated polysaccharides (SPs) from red algae have attracted considerable interest due to their antioxidant and anticancer properties. *Gracilaria corticata* is a rich source of sulfated polysaccharides with notable antioxidant and anticancer potential.

### Objectives

This study aimed to optimize the extraction process of SPs from *G. corticata,* determine their chemical and structural features, and evaluate their antioxidant and anticancer activities.

### Materials and methods

The SPs were extracted from *G. corticata* using ultrasound-assisted extraction. The chemical composition, molecular weight (230 kDa), and structure of the extracted SPs were characterized using standard biochemical assays and nuclear magnetic resonance (NMR) spectroscopy. Antioxidant capacity was evaluated using 2,2-diphenyl-1-picrylhydrazyl (DPPH) and 2,2'-azino-bis(3-ethylbenzothiazoline-6-sulfonic acid) (ABTS) radical scavenging assays. Cytotoxicity was assessed against HT-29 colon cancer cells and normal CCD-841 cells via 3-(4,5-dimethylthiazol-2-yl)-2,5-diphenyltetrazolium bromide (MTT) assay. Apoptosis induction was further examined through Annexin V-FITC/PI staining and gene expression analysis of apoptosis-related markers.

### Results

Ultrasound-assisted extraction was optimized at 12 min, 800 W power, and a solvent-to-biomass ratio of 30 mL/g, yielding 34.8% SPs. The SPs contained 84.4% carbohydrates, 8.7% sulfate, and 2.55% protein, with galactose identified as the

**Data availability statement:** All relevant data are within the paper and its Supporting information files.

**Funding:** The author(s) received no specific funding for this work.

**Competing interests:** The authors have declared that no competing interests exist.

predominant sugar. Molecular weight was determined to be 230 kDa. Antioxidant assays demonstrated dose-dependent activity, reaching 81.92% (DPPH) and 86.34% (ABTS) radical scavenging. SPs inhibited HT-29 cell viability with an $IC_{50}$ of 171 μg/mL and a selectivity index (SI) of 3.48. Apoptosis was induced via increased intracellular reactive oxygen species (ROS) and altered expression of key genes, including upregulation of *Bax*, *P53*, and *Caspase 3*, and downregulation of *BCL2*.

## Conclusions

SPs extracted from *G. corticata,* show strong antioxidant and selective anticancer activity through ROS-mediated apoptosis. These findings support their potential use as natural therapeutic agents in pharmaceutical development.

## 1. Background

Colorectal cancer (CRC) represents one of the most commonly diagnosed cancers, positioned third globally in terms of occurrence and second in terms of cancer-related mortality. In 2020, there were approximately 1.93 million newly identified cases and around 0.94 million cancer-related deaths reported worldwide [1]. Research indicates that polysaccharides play a vital protective role against intestinal damage caused by chemotherapy [2].

Recent research on marine macroalgae as a source of new bioactive compounds has opened promising avenues for cancer therapy [3]. Their renewable nature, non-toxicity, lack of side effects, and ease of cultivation make them ideal candidates for the development of natural therapeutic agents [4,5]. SPs extracted from marine red algae represent a structurally diverse class of biopolymers exhibiting a broad spectrum of bioactivities, including antioxidant, anticoagulant, and anticancer properties. Their chemical complexity, including varying degrees of sulfation and molecular weight, plays a crucial role in modulating these biological effects [6]. Despite promising bioactivities reported for SPs from various red algae, there remains limited mechanistic understanding and optimization of extraction methods specifically for *G. corticata*, necessitating further biochemical and pharmacological studies to fully realize their clinical potential. SPs from *Gracilaria corticata* exhibit notable antioxidant capacity and can induce apoptosis in cancer cells, making them significant candidates for anticancer therapy [7,8]. Given the global burden of colorectal cancer, which remains one of the leading causes of cancer mortality worldwide, natural compounds such as SPs from *Gracilaria corticata* have attracted growing attention due to their ability to induce ROS-mediated apoptosis selectively in cancer cells, suggesting considerable therapeutic potential [7–10].

The antioxidant actions of marine SPs are crucial, as they modulate cellular oxidative stress, which closely relates to cancer prevention [11,12]. Beyond their antioxidant action, SPs from various seaweed species have also been shown to induce apoptosis in cancer cells by influencing key signaling pathways, including *JNK*, *p38 MAPK*, and *caspase* cascades [13,14]. Due to the structural diversity of SPs among

algal species, their biological activities can vary significantly depending on their origin and composition. Therefore, establishing an efficient extraction strategy for recovering active SPs is essential, as it may yield non-toxic compounds with strong free-radical scavenging ability [15]. Observations from other marine-derived SPs further support their potential for selective cytotoxicity toward cancer cells.

The cytotoxic effects of SPs have been extensively documented across various cancer cell lines, including those of the skin, liver, breast, colon, and cervix [16]. In addition to their cytotoxicity, these compounds have been shown to suppress key oncogenic processes, including angiogenesis, metastasis, and immune evasion [17,18]. Although the underlying apoptotic mechanisms remain incompletely understood, specific SPs such as fucoidan have been reported to induce apoptosis by elevating intracellular ROS levels in cancer cells, thereby activating oxidative stress-mediated signaling pathways [17,19].

The coasts of Iran along the Persian Gulf showcase the greatest diversity and widest distribution of Rhodophyceae, particularly the species *G. corticata* [20]. These seaweeds represent a valuable local resource for the large-scale extraction of biologically active compounds. *G. corticata* was specifically selected for this study due to its distinct chemical profile of SPs, including high sulfate and galactan contents, unique patterns of galactose and xylose dominance, and elevated molecular weight, which together confer enhanced bioactive properties compared to other red algae [21]. Prior studies report that G. corticata's SPs outperform those of related species in antioxidant and selective anticancer activities [20]. Its abundance along the Persian Gulf coast not only allows for sustainable collection but also yields local SPs with unique structural features driven by environmental factors

## 2. Objectives

This study aims to optimize the ultrasound-assisted extraction of sulfated polysaccharides from *Gracilaria corticata*, thoroughly characterize their chemical composition and structural features, and evaluate their antioxidant capacities alongside selective anticancer activities against HT-29 colon cancer cells. Additionally, the research seeks to elucidate the underlying mechanism of apoptosis induction, focusing on reactive oxygen species (ROS) generation and gene expression modulation associated with apoptotic pathways.

## 3. Materials and Methods

### 3.1. Collection and identification of samples

Specimens of *G. corticata* were obtained from the seashore of the Qeshm island (Hormozgan province, Iran), northeastern Persian Gulf (56 ° 05'E/26°55'N) in June to August (during summer) 2022, when bioactive constituents are known to peak [22]. All specimens of *Gracilaria corticata* were initially identified in the field based on established morpho-anatomical criteria [23]. For taxonomic confirmation, representative voucher specimens were pressed, dried, and deposited in the official herbarium of the Department of Marine Biology at Islamic Azad University, Ahvaz. Approximately 5 kg of fresh seaweed biomass was harvested. The samples were washed, air-dried at room temperature, and milled to a fine powder, yielding approximately 0.5 kg, representing a fresh-to-dried powder ratio of 10:1. The living samples were collected by hand from the intertidal mudflat and transported to the laboratory. Samples were washed with tap water to remove extraneous substances (epiphytes, other algae, and invertebrates). Then, samples were washed with distilled water and air-dried at room temperature on absorbent paper. The dried samples were milled to a fine powder and stored at −20 ºC.

### 3.2. Extraction of SPs from G. corticata

The dried, milled powder of *Gracilaria corticata* was subjected to ultrasound-assisted extraction following protocols optimized from established literature [24]. Briefly, 9 g aliquots of dried alga were suspended in 600 mL of distilled water, and the mixture was heated at 50°C for 16 hours with gentle stirring. The resulting suspension was allowed to cool, and

insoluble debris was removed by centrifugation at 300g for 10 minutes. The supernatant was collected and subjected to ultrasonic irradiation (JP-010S, China) under variable durations and power settings according to the experimental design (Table 1). Following sonication, the extract was transferred to a chilled beaker. Absolute ethanol (99.5%, v/v) was added gradually until a final volume ratio of 3:1 (ethanol: extract) was achieved to precipitate polysaccharides. The mixture was incubated overnight at 4°C to maximize precipitation. The resulting crude polysaccharide precipitate was collected by centrifugation (3000g, 10 min), washed with cold ethanol, and re-dissolved in distilled water. Any remaining small molecular contaminants were removed by dialysis (molecular weight cut-off [MWCO] 8 kDa). The purified polysaccharide solution was freeze-dried to yield the final crude SPs [25]. Optimal extraction yield, composition, and molecular characteristics were determined in subsequent assays and in accordance with recently published methods for red algal polysaccharides. All steps were validated for reproducibility in triplicate batches. The extraction yield (%) of sulfated polysaccharides (SPs) was calculated using equation (15):

Extraction yield (%) = [Weight of dried SPs (g) / Weight of dried seaweed material (g)] × 100

### 3.3. Selective optimization of extraction yield

A central composite design (CCD) was applied for the optimization of sulfated polysaccharides yield by three independent variables, including ultrasonic wave irradiation time (A: min), ultrasonic power (B: W), and solvent to the raw material (C: %v/v). The design consisted of 17 runs (8 factorial runs, 4-star points, and 5 central runs). The extraction yield was recorded as a response. The software package Design Expert version 12 (Statsoft, USA) was used for the experimental design and analysis of the interactive effect of the variables. The experimental design matrix is presented in Table 1.

### 3.4. Chemical composition of SPs from G. corticata

The protein content of SPs was determined using the method of Lowry et al [26]. The sulfate dosage in sulfated polysaccharides was determined by the turbidimetric method as described in a previous study [27]. The total carbohydrate content of SPs was estimated by the phenol–sulfuric acid method, applying glucose as the standard [28]. The neutral monosaccharide composition of SPs was analyzed by an Agilent HPLC system (Agilent Technologies, Santa Clara, USA) equipped with an InertSustain C18 column (4.6 mm I.D. 250 mm) (GL Sciences Inc., Tokyo, Japan) according to the previous study described [29]. Determination of peak molar mass (Mpk) was done by the GPC with a Shimadzu (LC-20 CE) chromatograph coupled to a refractive index detector (RID-10A) applying a linear Polysep column (7.8 300 mm, cut-off: 107 g/mol) [30].

Monosaccharide reference standards (mannose, rhamnose, glucose, xylose, fructose) were used for qualitative and quantitative analysis. Calibration curves for each standard were constructed by plotting known concentrations against peak areas to ensure accurate monosaccharide quantification. Monosaccharide identification was performed by matching sample retention times with those of the authentic standards analyzed under the same chromatographic conditions. The chromatographic data were analyzed using dedicated software (AZUR version 5.0.10.0), and retention times were confirmed by reference to published literature values. Although no specific spectral library was used, identification was based

**Table 1. Experimental design levels.**

| Factors | Levels | | | | |
|---|---|---|---|---|---|
| | - α | −1 | 0 | 1 | α |
| A: Ultrasonic wave irradiation time (min) | 1.27 | 4 | 8 | 12 | 14.72 |
| B: Ultrasonic power (w) | 127.28 | 400 | 800 | 1200 | 1472.72 |
| C: Solvent to the raw material (v/v%) | 13.18 | 20 | 30 | 40 | 46.81 |

on standard retention time comparisons consistent with established protocols. For the estimation of polysaccharides in SPs, nuclear magnetic resonance (NMR) spectroscopy was used as proposed by Maciel et al [31].

### 3.5. Antioxidant activity of SPs from. corticata

**3.5.1. The assay of DPPH radical scavenging potential.** 2,2-Diphenyl-1-picrylhydrazyl (DPPH) radical scavenging assay was conducted following the method of Blois (1958), involving a stable free radical widely used for antioxidant measurement. Similarly, 2,2′-azino-bis(3-ethylbenzothiazoline-6-sulfonic acid) (ABTS) radical cation assay was employed to evaluate antioxidant capacity as described by Re et al. (1999) [32,33]. 1 mL of the SPs was added to 1 mL DPPH solution (0.2 mM in ethanol) and vigorously stirred. The mixture was incubated in the dark for 30 min at 25°C. Then, the absorbance of the solution was measured at 517 nm against a blank. The more the absorbance of the reaction mixture decreased, the stronger the DPPH scavenging ability. The ability to scavenge DPPH was calculated by the following formula:

$$\text{DPPH scavenging ability (\%)} = (1 - A_1 - A_2/A_0) \times 100$$

$A_0$: absorbance of the control solution only without the sample. $A_1$: absorbance of the test sample mixed with DPPH solution. $A_2$: absorbance of the blank.

**3.5.2. ABTS radical scavenging capacity.** The ABTS radical scavenging capacity of SPs was measured using the previously described method [34]. In brief, for preparing the ABTS radical cation solution, 7 mM of ABTS solution was mixed with 2.45 mM potassium persulfate and placed in the dark at room temperature for 16 h. Then, the prepared ABTS-+ solution (7 mM ABTS and 2.45 mM potassium persulfate) was diluted with methanol until its absorbance at 734 nm reached 1.1 ± 0.02 units. After that, 2.7 mL ABTS in various concentrations was added to 2 mL SPs and kept in the dark at room temperature for 30 min. The adsorption of the solution was read at 734 nm by a spectrophotometer. The free radical scavenging activity of ABTS was calculated according to the following equation:

$$\text{ABTS scavenging ability (\%)} = (1 - A_1 - A_2 / A_0) \times 100$$

$A_0$: absorbance of the control solution only without the sample. $A_1$: absorbance of the test sample mixed with DPPH solution. $A_2$: absorbance of the blank.

### 3.6. Evaluation of Anticancer Activity of SPs from Gracilaria corticata

**3.6.1. MTT cell viability assay.** An MTT stock solution was prepared at a concentration of 5 mg/mL in phosphate-buffered saline (PBS), sterilized, and stored at 4°C until use. For cell viability assessment, 10 μL of the MTT stock solution was added to each well containing 100 μL of culture medium, resulting in a final concentration of approximately 0.45 mg/mL. Plates were incubated for 3 hours at 37°C to allow formazan crystal formation. The Research and Production Complex Pasteur Institute of Iran provided the normal human colon cell line CCD- 841 and the colon cancer cell line HT-29. A 96-well microplate was used to seed cell lines. Dulbecco's modified Eagle's medium Dulbecco's modified Eagle's medium (DMEM) with 10% FBS was used to cultivate the microplate containing HT29 and CCD-841 cells. It was then incubated for 12 h at 37 °C with 5% $CO_2$. The MTT assay was used to evaluate the viability of HT29 and CCD-841 cells following a 72 h treatment with varying concentrations of SP. In 96-well microplates, cells were seeded in triplicate with a density of $5 \times 10^3$ cells per well. Following an overnight incubation period, the cells were subjected to 72 h of SP treatment at concentrations of 0, 12, 5, 25, 50, 100, 200, and 200 μM. Each well was then filled with the MTT solution, and the plates were incubated for 4 h at 37 °C to enable viable cell dehydrogenases to convert the MTT to insoluble formazan. After removing the supernatant, the intracellular formazan was dissolved for 15 minutes at room temperature in 100 μL/well of dimethyl sulfoxide (DMSO). A microplate reader was used to measure the absorbance at 570 nm (Dynatech Laboratories, Chantilly, VA, USA). To determine the percentage of cells that were viable, the formula was as follows: (Mean absorbance

in test wells / Mean absorbance in control wells) × 100. Each cell line's survival curve was then plotted according to the relationship between SP concentrations and the quantity of surviving cells [35,36]. Concentration-response curves must be created for every dosage in order to determine the 50 percent inhibitory concentration ($IC_{50}$). The selectivity index (SI), which can be computed using the formula SI $IC_{50}$ (non-cancer cells)/$IC_{50}$ (cancer cells), was used to assess the cytotoxic selectivity of SP. A high degree of selectivity is indicated by an SI larger than 2 [37]. Three groups were created from the HT-29 cells. Cells were distributed into three groups: one group received no treatment and served as the control (untreated HT-29 cells), while the other two groups were treated with 1/2 $IC_{50}$ and 1/4 $IC_{50}$ concentrations of SPs, respectively. All treatments were performed in triplicate when the cells reached 70–80% confluence. The cells were kept at 37 °C and 95% humidity for 24 h in a $CO_2$ incubator.

**3.6.2. Intracellular reactive oxygen species estimation.** The fluorescent probe 20,70-dichlorofluorescein diacetate (DCFDA) was used in a previously reported procedure to quantify intracellular reactive oxygen species (ROS) [38]. After the cells were treated, they were exposed to 25 μM $H_2O_2$ for 2 h. Following a PBS wash, the cells were treated with DCFDA (5 μM) and incubated for 30 minutes at 37°C in the dark. A fluorescence microplate reader was used to measure the fluorescence intensity of DCFDA after another round of PBS washing. A percentage of the control was used to represent the intracellular ROS levels.

**3.6.3. Cell apoptosis assay.** According to the supplier's instructions, an AnnexinV-FITC/PI apoptosis kit (BioLegend, San Diego, CA) was used to detect the apoptosis of the cell line HT-29. In short, 100 μL of Annexin-binding buffer was used to resuspend the viable cells ($1 \times 10^5$) that were collected from experimental groups after they had been rinsed with a cell staining buffer (BioLegend). After 15 minutes of incubation with 2.5 μL Annexin V-FITC, the suspension cells were incubated for 10 minutes at room temperature in the dark with 2.5 μL propidium iodide (PI). Using the proper filters, a flow cytometer (BD FACSCaliburTM, USA) identified macrophages that were FITC or PI positive.

**3.6.4. Gene expression analysis.** Following RT-PCR analysis, total RNA was extracted from the cells using the Trizol reagent (Qiagen RNA extraction method). Nanodrop and 1% gel electrophoresis were used to measure the RNA's concentration and integrity, respectively. Reverse transcription was performed using RevertAid H Minus Reverse Transcriptase to measure the expression levels of the *Bax*, *Bcl2*, *P53*, and *caspase 3* genes. This was followed by qPCR with 2X SYBR Green Master Mix and particular primers (Table 2). Thermo Scientific (Waltham, Massachusetts, USA) provided all of the qPCR kits. Two μL of cDNA, 1 μL of each primer, and twelve to 5 μL of Maxima SYBR Green Master Mix made up the 25 μL qPCR mixture. After a 10 min initial step at 95°C, there were 45 cycles of denaturation at 95°C for 15 s, annealing at 60°C for 30 s, and extension at 72°C for 30 s. Calculated using the 2 - $\Delta\Delta C_t$ method, the relative expression of the target genes was normalized to the housekeeping gene *glyceraldehyde-3-phosphate dehydrogenase (GAPDH)* [39].

## 3.7. Statistical analysis

The data were analyzed using the one-way ANOVA followed by Newman–Keuls tests. Differences were considered statistically significant at $P < 0.05$. All the experiments were performed in triplicate and presented as a mean ± standard deviation [32]. All statistical analysis was done in IBM SPSS Statistics for Windows, version 27 (IBM Corp., Armonk, N.Y., USA).

**Table 2. Gene primer sequences applied in qPCR.**

| Gene | Forward | Reverse |
|------|---------|---------|
| *Bcl2* | 5'TTGATGGGATCGTTGCCTTATGC3' | 5'CAGTCTACTTCCTCTGTGATGTTG3' |
| *Bax* | 5'GGACGAACTGGACAGTAACATGG3' | 5'GCAAAGTAGAAAAGGGCGACAAC3' |
| Caspase3 | 5'GAAGCGAATCAATGGACTCTGG3' | 5'GACCGAGATGTCATTCCAGTGC3' |
| *P53* | 5'TAACAGTTCCTGCATGGGCGGC3' | 5'AGGACAGGCACAAACACGCACC3' |
| *GAPDH* | 5'GGTGAAGGTCGGAGTCAACG3' | 5'TGAAGGGGTCATTGATGGCAAC3' |

 

## 4. Results

### 4.1. Response surface experiment results

The effect of ultrasonic wave irradiation time, ultrasonic power, and solvent on the raw material on the extraction yield of SPs was examined by CCD in levels. Analysis of the independent variables revealed that the extraction yield of the SPs could be described by a linear and 2FI model (Table 3).

Based on the response surface plot and its contour plot of the extraction yield of the SPs, the interaction of ultrasonic wave irradiation time and ultrasonic power with the extraction yield response was significant. However, the interaction of ultrasonic power and solvent to the raw material with the extraction yield response was not significant (Fig 1). According to the software analysis, the model-predicted optimal extraction yield of the SPs was 27.8% under conditions of ultrasonic time of 12 min, ultrasonic power of 849.44 W, and solvent to dry matter ratio of 23.31 mL/g.

### 4.2. Analyzing data and fitting models

Quadratic response surface models were developed to predict the yield of sulfated polysaccharides ($Y_{yield}$) based on multiple regression analysis of experimental data, incorporating ultrasonic extraction parameters (A, B, and C).

The Regression Equation for SPs Yield from *G. corticata* via Ultrasonic Extraction

$$Y_{yield} = 1.514 + 0.005\,A + 0.402\,B + 0.112\,C - 0.00008\,AB + 0.00004\,AC + 0.004\,BC - 3.51761E\text{-}06\,A^2 - 0.017\,B^2 - 0.003\,C^2$$

### 4.3. Chemical composition of SPs

**4.3.1. Molecular weight.** According to GPC analysis, the highest molecular weight was obtained in 12.22 min with 670 kDa (Table 4). The average polysaccharide weight was 230 kDa.

**4.3.2. Chemical composition.** Carbohydrates (84.40%) and sulfate (8.7%) were the highest compounds in SPs (Table 5).

**4.3.3. Monosaccharide composition.** The HPLC profile of the standards, including mannose, rhamnose, glucose, xylose, fructose, and extracted SPs, is presented in Fig 2. The retention times (Rt) and relative concentrations of monosaccharides identified in the sulfated polysaccharides from *Gracilaria corticata* are summarized in Table 6, based on calibration and comparison with authentic standards. The major sugar in the SPs was galactose (40.14%). The remaining

**Table 3. The variance analysis of the main responses of the extraction yield of the SPs from red algae (*G. corticata*).**

| Source | Sum of Squares | df | Mean Square | F-value | p-value | |
|---|---|---|---|---|---|---|
| Model | 9.91 | 9 | 1.10 | 30.16 | 0.0001> | significant |
| A-Ultrasonic power | 0.6250 | 1 | 0.6250 | 17.13 | 0.0044 | |
| B-Extraction time | 4.75 | 1 | 4.75 | 130.90 | 0.0001> | |
| C- Solvent to the raw material | 0.1124 | 1 | 0.1124 | 3.08 | 0.1227 | |
| AB | 0.1378 | 1 | 0.1378 | 3.78 | 0.0931 | |
| AC | 0.208 | 1 | 0.2080 | 5.70 | 0.0483 | |
| BC | 0.2145 | 1 | 0.2145 | 5.88 | 0.0458 | |
| A² | 0.8487 | 1 | 0.8487 | 23.26 | 0.0019 | |
| B² | 0.2219 | 1 | 0.2219 | 60.08 | 0.0431 | |
| C² | 0.2622 | 1 | 0.2622 | 7.18 | 0.0315 | |
| Residual | 0.2554 | 7 | 0.0365 | | | |
| Lack of Fit | 0.2482 | 5 | 0.0495 | 13.79 | 0.069 | not significant |
| Pure Error | 0.0072 | 2 | 0.0036 | | | |
| Cor Total | 10.16 | 16 | | | | |

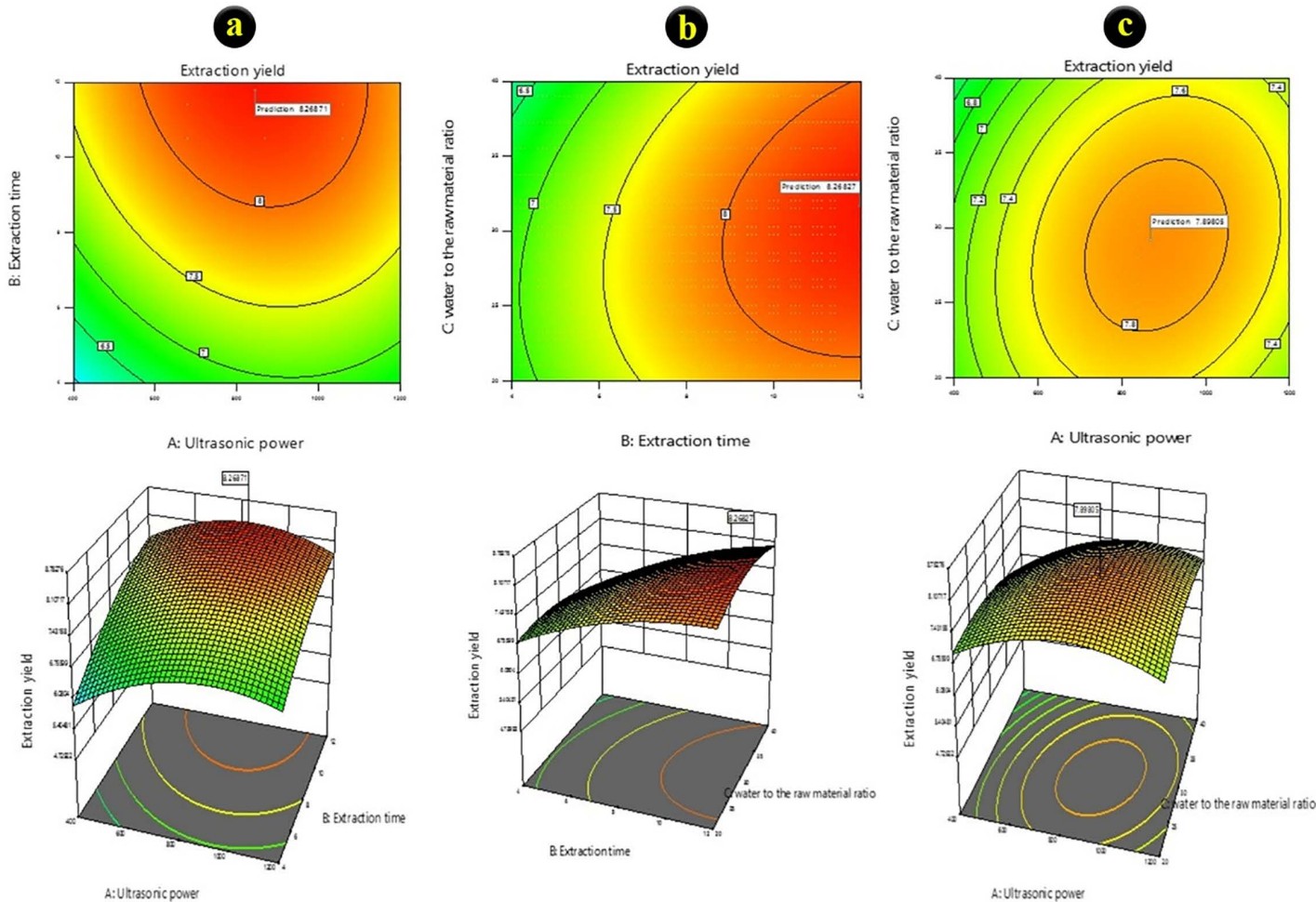

**Fig 1. Contour plot and its response surface plot of the extraction yield of the SPs from red algae (*G. corticata*).** a) the interaction of ultrasonic wave irradiation time with extraction yield response, b) the interaction of solvent to the raw material with extraction yield response, and c) the interaction of ultrasonic power with extraction yield response.

**Table 4. The molecular weight of the extracted SPs from red algae (*G. corticata*) according to the extraction time.**

| Time (mine) | Molecular weight (KDa) |
|---|---|
| 22.12 | 670 |
| 30.71 | 410 |
| 38.27 | 270 |
| 43.74 | 150 |
| 47.05 | 80 |
| 49.11 | 25 |
| 50.09 | 5 |

                                      

**Table 5. Chemical composition of SPs extracted from red algae (*G. corticata*) according to the extraction time (mean± SD).**

| Composition | Percentage |
|---|---|
| carbohydrate | 84.40±1.12 |
| Protein | 2.55±0.09 |
| Sulphate | 8.70±0.06 |

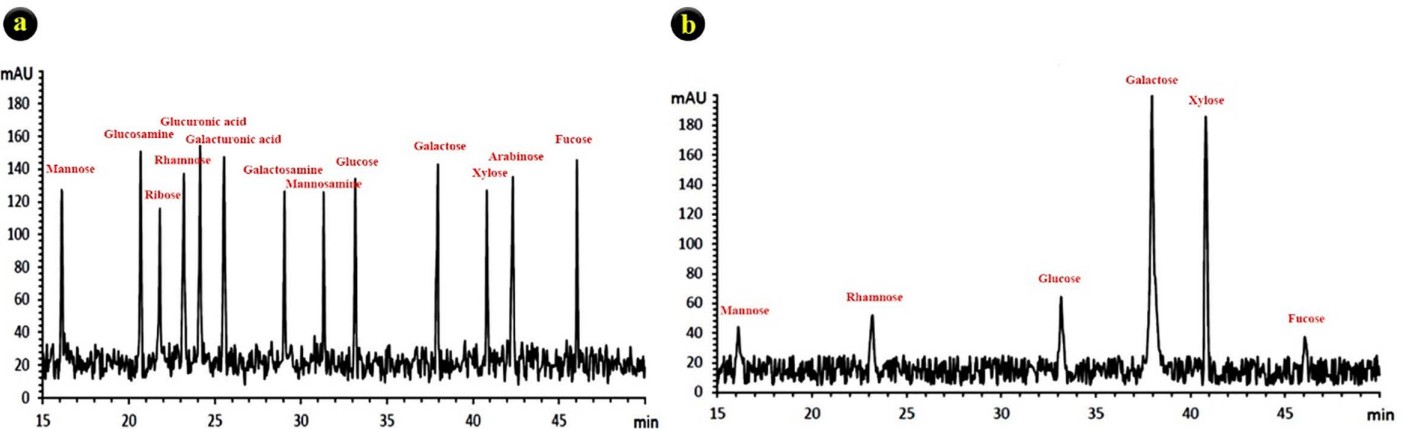

**Fig 2. Monosaccharide composition of the SPs from red seaweed (*G. corticata*).** a) Standard, b) Sample.

**Table 6. Retention times (Rt, minutes) and relative concentrations (%) of monosaccharides identified in sulfated polysaccharides extracted from *Gracilaria corticata* by HPLC analysis.**

| Monosaccharide | Retention Time (Rt, min) | Concentration (% of total) |
|---|---|---|
| Galactose | 8.5 | 40.14 |
| Xylose | 7.5 | 37.75 |
| Glucose | 6.3 | 7.32 |
| Rhamnose | 9.0 | 5.83 |
| Mannose | 10.2 | 5.08 |
| Fructose | 5.4 | 3.75 |

mono-sugars include fructose (3.75%), mannose (5.08%), rhamnose (5.83%), glucose (7.32%), and xylose (37.75%), which were minimum amounts compared to the standard mono-sugars (Fig 2).

In the present study, the polysaccharide exhibited a lower galactose content and higher levels of glucose.

**4.3.4. NMR analysis of SPs.** The NMR analysis (The $^1$H and $^{13}$C NMR spectra) to investigate the red seaweed *G. corticata* polysaccharide structure was presented in Fig 3. The $^1$H NMR spectrum is somewhat complex (Fig 3a). The signals from the α anomeric proton at δ 7.34 and 8.29 were assigned to 3,6 α-l-anhydrogalactose (LA) and α-l-galactose-6-sulfate (L-6S), respectively. The anomeric region of $^{13}$C-NMR revealed four main signals based on literature data [31,40], including C-1 from -d-galactose linked to -l-galactose-6-sulfate at 29.74; C-1 of -d-galactose linked to

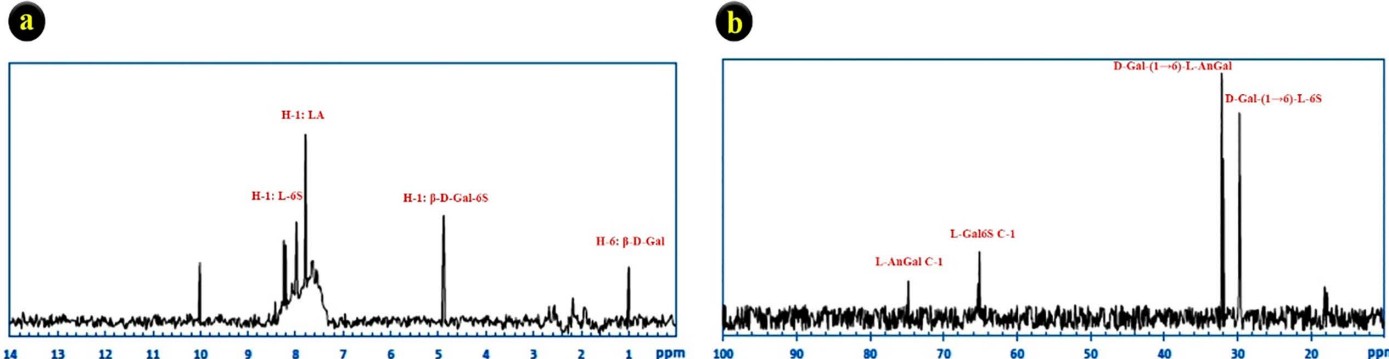

**Fig 3. NMR spectra of the SPs from red seaweed (*G. corticata*) in D₂O.** a) H NMR spectrum, b) C NMR spectrum.

3.6-L-anhydrogalactose at 31.95; C-1 of the −1-galactopyranose-6-sulfate moiety at 65.13; and C-1 of 3,6-anhydro-1-galactopyranose at 74.81 (Fig 3b).

### 4.4 Antioxidant activity

**4.4.1. DPPH free radical scavenging assay.** The use of DPPH free radical as a stable free radical has been widely accepted to estimate the free radical scavenging activities of antioxidants. The inhibitory effect of SPs from red seaweed (*G. corticata*) on DPPH radical was concentration dependent, in which 200–3000 µg/mL showed 41.11%–81.92% of inhibition, respectively (Fig 4). The highest scavenging effect (81.92%) was observed at the concentration of 3 mg/mL of SPs. The results revealed a constant increase in DPPH antioxidant activity with the increased concentration.

**4.3.2. ABTS free radical scavenging assay.** In the present study, the SPs revealed significant ABTS scavenging ability in a concentration-dependent manner. The scavenging ability for the various concentrations of 0.2–3 mg/mL was 43.92%−86.34% (Fig 4). The maximum scavenging ability (81.92%) was observed at the concentration of 3 mg/mL of

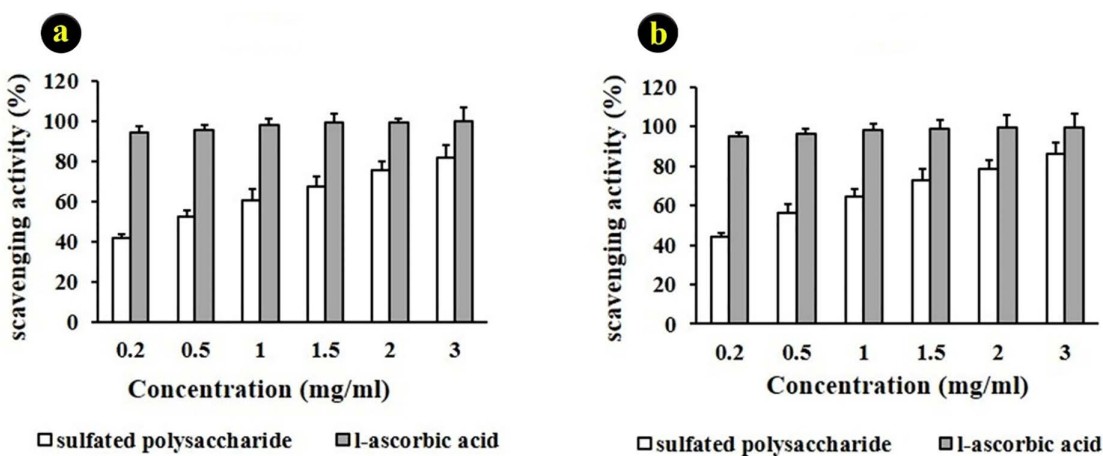

**Fig 4. Antioxidant activity of SPs from red seaweed (*G. corticata*).** a) Scavenging DPPH radicals b) ABTS scavenging assay. (Mean ±SD), Different letters in each group indicate a significant difference (P < 0.05), (n = 6).

SPs. The ABTS scavenging ability of the SPs in this study was higher than the SPs from *M. oxyspermum* (76.81%) at a concentration of 125 µg/mL and *G. corticata* (74.5%) at a concentration of 125 µg/mL.

### 4.5. Correlation analysis

The correlation between antioxidant activity and the chemical composition, including the contents of carbohydrates, sulfates, proteins, and sugar of the SPs, was presented in Table 7.

A negative significant correlation between carbohydrate content and scavenging activity against DPPH and ABTS was observed, which suggested that the carbohydrate content might promote the scavenging effect. In the present study, a significant negative correlation between sulfate content and the antioxidant effect was observed (Table 6). A highly positive correlation was found between the protein content and the antioxidant effect (Table 6).

### 4.6. SPs as anti-cancer agents

Cell viability assays revealed that treatment with the extracted SPs did not significantly affect the survival of CCD-841 cells. In contrast, it markedly reduced the viability of HT-29 cells, with more than 85% of cells rendered non-viable at a concentration of 400 µg/ mL. The $IC_{50}$ value for HT-29 cells was determined to be $171 \pm 3$ µg/ mL (Fig 5a, 5b). The SI for SP against HT-29 cells after 72 h of treatment was calculated as 3.48, indicating strong selectivity for this cell line.

Exposure of HT-29 cells to sublethal doses of SP significantly increased intracellular ROS levels, with the highest ROS content observed in the ½ $IC_{50}$ group. The relative fluorescence percentages for ROS were as follows: control ($98 \pm 1.9\%$), ½ IC50 ($328 \pm 2\%$), and ¼ IC50 ($53 \pm 2.7\%$). Apoptosis analysis revealed that treatment with ½ $IC_{50}$ SP resulted in the highest proportion of early and late apoptotic cells ($P < 0.05$; Fig 5c, 5d).

Treatment of HT-29 cells with SP revealed significant changes in gene expression. Specifically, the expression levels of *Bax*, *P53*, and *Caspase 3* were maximally elevated following treatment with ½ $IC_{50}$ SP, compared to other groups ($P < 0.05$) (Fig 6). In contrast, the expression of the *BCL2* gene exhibited the opposite trend, showing a maximum reduction under the same treatment conditions ($P < 0.05$) (Fig 6).

The selective cytotoxicity of SPs toward cancer cells, with minimal impact on normal cells, is a critical feature for their potential therapeutic use. In this study, the $IC_{50}$ value of 171 µg/mL for HT-29 cells, combined with negligible effects on CCD-841 cells, highlights their promising anticancer selectivity. The elevated intracellular ROS levels observed following SP treatment, particularly at ½ $IC_{50}$ concentration, highlight oxidative stress as a central component of their anticancer mechanism. The increased ROS levels likely trigger the activation of *P53*, which in turn regulates the expression of key apoptotic effectors. This is evidenced by the observed upregulation of pro-apoptotic genes (*Bax*, *P53*, and *Caspase 3*)

**Table 7. Pearson's correlation coefficients (r) of antioxidant activity against DPPH and ABTS with sulfate, carbohydrate, and protein of SPs from red seaweed (*G. corticata*) (\* Correlation is significant at p ≤ 0.05.).**

| Antioxidant Assay | Protein | Carbohydrate | Sulfate |
|---|---|---|---|
| DPPH | −0.658* | −0.932* | not determined |
| ABTS | 0.985* | −0.911* | −0.960* |
| **Chemical Component** | **Protein** | **Carbohydrate** | **Sulfate** |
| Protein | 1.000 | −0.932* | 0.958* |
| Carbohydrate | −0.932* | 1.000 | 0.953* |
| Sulfate | 0.958* | 0.953* | 1.000 |
| Protein | 1.000 | −0.932* | 0.958* |

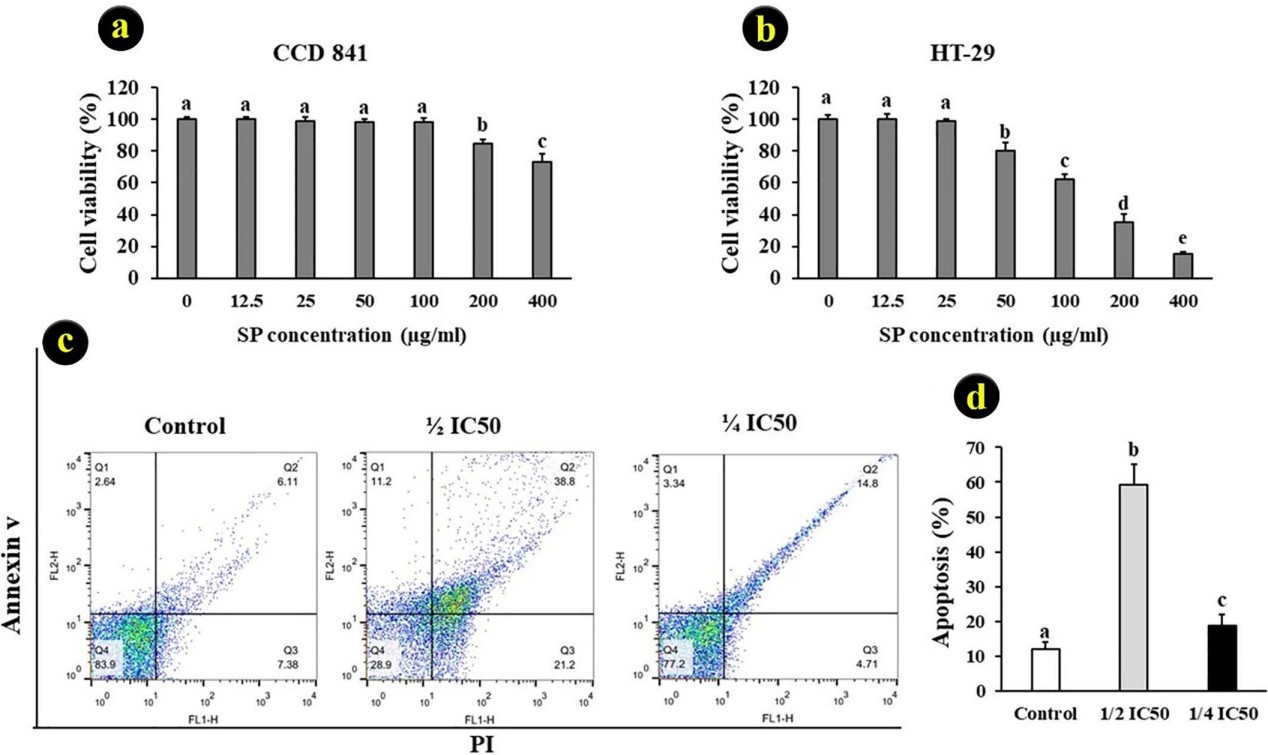

**Fig 5. Cytotoxic activity of SPs from red seaweed (*G. corticata*) against CCD- 841.** a) and HT-29 b) Histogram plot of apoptotic HT-29 cells after 72 h treatment with SP. c) Histogram plot was quantified by flow cytometry, and the HT-29 cells were stained with Annexin V-FITC and propidium iodide (PI). d) Different letters in each group indicate a significant difference (P < 0.05) (mean ± SD).

alongside the downregulation of the anti-apoptotic gene *BCL2*, indicating the engagement of the intrinsic (mitochondrial) apoptotic pathway.

## 5. Discussion

This study evaluated the extraction, structural features, and biological activity of SPs obtained from *G. corticata* to elucidate their potential as natural therapeutic agents. Using an ultrasound-assisted extraction protocol optimized based on model predictions, an experimentally verified yield of 34.8% was obtained under ideal conditions, which exceeds the model-predicted yield of 27.8% and compares favorably with yields previously reported for other *Gracilaria* species [41,42] but is lower than the 63% observed in *G. debilis* [43]. Variations in SP yield are likely due to differences in extraction parameters, seaweed species, and environmental factors [21,44]. In addition to affecting extraction yield, ultrasonic parameters can significantly influence the molecular weight distribution of extracted polysaccharides. Previous studies have demonstrated that ultrasonic power modulates cavitation intensity, generating localized shear forces capable of fragmenting glycosidic linkages and reducing molecular weight [45]. Similarly, solvent-to-material ratio impacts mass transfer efficiency and may alter chain stability during extraction [46]. The variation in molecular weight observed in the present study is therefore likely attributable to the combined effects of ultrasonic time, power, and solvent conditions.

The SPs are critical determinants of biological activity [21,44]. The monosaccharide composition identified in this study differs from previous reports on *G. corticata*, where galactose content exceeded 90% [21]. Recent structural studies have revealed that not only the total sulfate content but also the specific sulfation pattern on galactose residues significantly

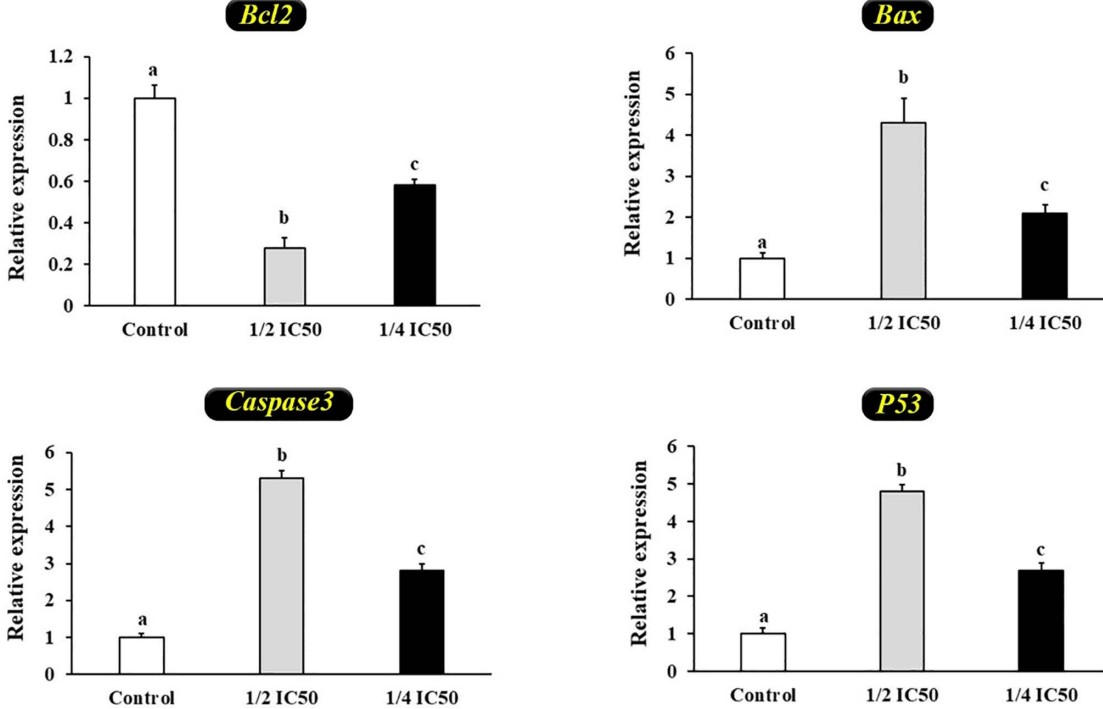

**Fig 6. The expression levels of apoptotic genes (*Bcl2*, *Bax*, *Caspase3*, and *P53*) in HT-29 cells were analyzed following treatment with SP from the red seaweed *G. corticata* at ½ IC$_{50}$ and ¼ IC$_{50}$ concentrations.** Gene expression data were normalized to the housekeeping gene GAPDH and are presented as mean fold change±SD. Each experiment was conducted in triplicate, with a total of three independent experiments (n=6). Significant differences between groups, indicated by different letters, were determined at P<0.05.

modulate bioactivity [47,48]. For example, 2,3-di-sulfated galactose units have been shown to enhance antioxidant and anticancer efficacy by improving interactions with reactive oxygen species and cancer cell receptors. These sulfation motifs influence molecular conformation, facilitating targeted cellular responses. Thus, the bioactivity observed in our SPs is likely attributable to both the abundance and precise structural arrangement of sulfate groups along the galactan backbone, consistent with emerging structure-activity relationships in marine-derived polysaccharides. Chemical characterization revealed that the SPs consisted of 84.4% carbohydrate, 8.7% sulfate, and 2.55% protein, indicating a galactan-rich and moderately sulfated structure, which is consistent with previous findings on *G. corticata* [21,44]. Compared to other species, such as *Codium tomentosum*, which lacks protein and contains over 91% carbohydrate [49], the moderate protein content in the current study may contribute to its antioxidant potential via amino side groups acting as electron donors [50]. The significant correlations observed between sulfate, protein content, and antioxidant activity are consistent with previous studies demonstrating that sulfate groups enhance radical scavenging through increased electron-donating capacity [50,51]. Additionally, amino-containing moieties may contribute to antioxidant activity by altering molecular polarity and facilitating hydrogen donation [50]. These findings further support the importance of structural features in determining the bioactivity of marine sulfated polysaccharides. The carbohydrate content of 84.4% observed in our SPs aligns closely with previous studies analyzing *G. corticata* polysaccharides, although sulfate content shows some variation across studies [21]. Another study reported that the SPs of *Codium tomentosum* contained 91.6% of carbohydrates and 0% of protein [49], which has a higher carbohydrate content compared to this study. This variation may be attributed to differences in extraction methods and environmental factors influencing polysaccharide composition.

The SPs displayed a distinct monosaccharide composition, with galactose (40.14%) and xylose (37.75%) as dominant sugars. This pattern differs from prior reports on *G. corticata*, where galactose accounted for over 90% [21], and from Mastocarpus stellatus, which showed over 95% galactose [52]. Such differences may influence solubility, viscosity, and ultimately biological activity. The lower galactose and higher xylose content observed here could indicate structural branching, which may enhance radical scavenging by increasing surface interactions.

Molecular weight analysis via GPC revealed that extraction time significantly impacted SP size, with the highest molecular weight (670 kDa) observed at 12.22 minutes. The average molecular weight was 230 kDa, substantially higher than the 31.5 kDa reported for SPs from *G. lemaneiformis* [53]. High molecular weight is generally associated with more potent antioxidant and immunomodulatory effects, likely due to enhanced steric hindrance and interaction with reactive species [30].

NMR spectroscopy confirmed the structural complexity of the extracted SPs. The ¹H-NMR spectrum exhibited signals at δ 7.34 and 8.29 ppm, corresponding to 3,6-anhydro-α-L-galactose (LA) and α-L-galactose-6-sulfate (L-6S), respectively. The ¹³C-NMR data supported the presence of sulfated galactopyranose and anhydrogalactose linkages, consistent with a repeating agar-like disaccharide backbone [31,40]. These sulfated motifs are known to enhance bioactivity by improving solubility, increasing charge density, and facilitating interactions with ROS or cancer cell membranes [15,16].

Biologically, the SPs demonstrated strong antioxidant capacity, with maximum DPPH and ABTS scavenging activities of 81.92% and 86.34%, respectively. These values exceed those of similar SPs from *M. oxyspermum* (76.81%) and *C. officinalis* (15.2%) [54,55]. Correlation analysis revealed significant relationships between antioxidant activity and the content of sulfate and protein. High sulfate content has previously been linked to improved electron-donating ability and radical neutralization, while the presence of amino groups may enhance polarity and reactivity [50,51].

The SPs also exhibited selective cytotoxicity toward HT-29 colon cancer cells, with minimal toxicity against normal CCD-841 colon cells. The $IC_{50}$ value of 171 µg/mL and SI of 3.48 demonstrate a favorable therapeutic window. This supports previous observations where SPs from red seaweeds induced cytotoxicity in cancer cells while sparing normal tissues [56–58].

Mechanistically, treatment with SPs led to significant increases in intracellular ROS levels, particularly at a ½ $IC_{50}$ concentration, suggesting that oxidative stress is a key trigger for apoptosis. Expression analyses confirmed the upregulation of *Bax*, *P53*, and *Caspase 3*, and the downregulation of *Bcl2*, indicating activation of the mitochondrial apoptotic pathway. These findings align with reports on other marine SPs that trigger ROS-mediated apoptosis via *P53* signaling [19,59,60]. Additionally, studies have shown that SPs may influence upstream regulators such as *JNK* and *p38 MAPK* pathways, which further modulate cell death and proliferation [8,13].

Taken together, these findings underscore the dual functional potential of *G. corticata*-derived SPs: acting as antioxidants under physiological conditions and as pro-oxidants in cancer cells. This paradoxical effect is advantageous, as it capitalizes on the higher oxidative vulnerability of cancer cells compared to normal cells [9]. The structural elements identified in the NMR and compositional analyses may be directly responsible for this selective behavior, highlighting the importance of structure–activity relationships in marine bioactives.

Nevertheless, the study has some limitations. The biological assessments were conducted exclusively in vitro, and the lack of *in vivo* validation restricts insight into pharmacokinetics, bioavailability, and systemic toxicity. Furthermore, testing was limited to a single cancer cell line. Future research should expand to multiple cancer models and evaluate potential synergistic effects with standard chemotherapeutics. Detailed proteomic and metabolomic analyses could also provide deeper insights into the mechanisms.

Regarding the scalability and practicality of ultrasound-assisted extraction (UAE) for industrial applications, UAE offers several advantages, such as increased extraction efficiency, reduced solvent consumption, shorter extraction times, and lower energy usage compared to conventional extraction methods [61]. These features align with environmental and cost-saving goals desirable in large-scale production. However, scaling up the UAE poses challenges, including

non-uniform cavitational energy distribution in larger vessels, attenuation of ultrasonic waves, and equipment design complexities to achieve consistent power delivery across substantial volumes [62]. Strategies such as multi-transducer reactors, continuous flow ultrasonic systems, and integration with other extraction technologies are under investigation to overcome these limitations and improve throughput. Therefore, while the UAE shows promising industrial applicability for extracting bioactive compounds from seaweeds, further optimization and engineering advances are required to fully realize its commercial potential.

## 6. Conclusion

This study optimized ultrasound-assisted extraction of sulfated polysaccharides from Gracilaria corticata, achieving a yield of 34.8% under defined conditions (12 min, 800 W, 30 mL/g). The extracted polysaccharides were characterized by high carbohydrate content (84.4%), moderate sulfation (8.7%), and a molecular weight of 230 kDa, with galactose and xylose as dominant monosaccharides. Functionally, the SPs exhibited strong dose-dependent antioxidant activity and selective cytotoxicity toward HT-29 colon cancer cells ($IC_{50}$ = 171 µg/mL; SI = 3.48). Mechanistic analyses demonstrated that apoptosis induction was mediated by increased intracellular ROS levels, upregulation of Bax, P53, and Caspase-3, and downregulation of Bcl2, confirming activation of the intrinsic apoptotic pathway. Collectively, these findings establish a clear structure–activity relationship and demonstrate the potential of *G. corticata*-derived sulfated polysaccharides as selective anticancer agents with dual antioxidant and pro-oxidant functionality.

## Author contributions

**Conceptualization:** Zeynab Falihzadeh.

**Data curation:** Zeynab Falihzadeh.

**Formal analysis:** Zeynab Falihzadeh.

**Funding acquisition:** Zeynab Falihzadeh.

**Investigation:** Zeynab Falihzadeh.

**Methodology:** Laleh Roomiani.

**Project administration:** Laleh Roomiani.

**Resources:** Laleh Roomiani.

**Software:** Laleh Roomiani.

**Supervision:** Laleh Roomiani.

**Validation:** Laleh Roomiani.

**Visualization:** Laleh Roomiani.

**Writing – original draft:** Laleh Roomiani.

**Writing – review & editing:** Laleh Roomiani.

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
