## [Decision Letter · Decision Letter 0]

18 Sep 2025

PONE-D-25-45707Targeting colon cancer through apoptosis: the antioxidant and anticancer potential of sulfated polysaccharides from Gracilaria corticatePLOS ONE

Dear Dr. roomiani,

Thank you for submitting your manuscript to PLOS ONE. After careful consideration, we feel that it has merit but does not fully meet PLOS ONE’s publication criteria as it currently stands. Therefore, we invite you to submit a revised version of the manuscript that addresses the points raised during the review process.

We look forward to receiving your revised manuscript.

Kind regards,

Fahrul Nurkolis

Academic Editor

PLOS ONE

Journal Requirements:

4. Please upload a new copy of Figure 1 as the detail is not clear. Please follow the link for more information:

https://blogs.plos.org/plos/2019/06/looking-good-tips-for-creating-your-plos-figures-graphics/

https://blogs.plos.org/plos/2019/06/looking-good-tips-for-creating-your-plos-figures-graphics/

**Additional Editor Comments:**

Dear Authors, Thank you for submitting your manuscript to our journal. After careful consideration, I must inform you that your paper is not yet ready for publication in its current form.

Based on the reviewers’ evaluations, I strongly recommend that you revise the manuscript thoroughly in accordance with the suggestions provided by all seven reviewers. In particular, please note that Reviewers 4 and 6 have recommended rejection, and their concerns should be carefully addressed and taken into account.

We encourage you to carefully consider all feedback and submit a revised version that reflects the necessary improvements. I look forward to receiving your revision.

Reviewers' comments:

Reviewer's Responses to Questions

**Comments to the Author**

1. Is the manuscript technically sound, and do the data support the conclusions?

Reviewer #1: Yes

Reviewer #2: Yes

Reviewer #3: Partly

Reviewer #4: No

Reviewer #5: Yes

Reviewer #6: Partly

Reviewer #7: Yes

2. Has the statistical analysis been performed appropriately and rigorously? 

Reviewer #1: Yes

Reviewer #2: Yes

Reviewer #3: Yes

Reviewer #4: Yes

Reviewer #5: Yes

Reviewer #6: No

Reviewer #7: Yes

3. Have the authors made all data underlying the findings in their manuscript fully available?

Reviewer #1: Yes

Reviewer #2: Yes

Reviewer #3: Yes

Reviewer #4: No

Reviewer #5: Yes

Reviewer #6: Yes

Reviewer #7: Yes

4. Is the manuscript presented in an intelligible fashion and written in standard English?

Reviewer #1: No

Reviewer #2: Yes

Reviewer #3: Yes

Reviewer #4: No

Reviewer #5: Yes

Reviewer #6: No

Reviewer #7: Yes

5. Review Comments to the Author

Reviewer #1: This manuscript entitled “Targeting colon cancer through apoptosis: the antioxidant and anticancer potential of sulfated polysaccharides from Gracilaria corticate”, explores the contribution of marine-derived bioactive compounds for potential pharmaceutical applications. The manuscript is publishable with some refinement as there are several structural, language, and scientific issues that need attention before publication.

Reviewer #2: Though the mansucript is well written, there are major issues that needs to addresed. Please see the attached reviwer report for further details and comments. Address those issues in detial and resubmit the mansucript.

Reviewer #3: 1. There are some grammatical, alignments and typographical errors are noted in the manuscript and it should be thoroughly checked and corrected throughout the manuscript.

2. The use of abbreviations in the abstract may distract readers who wish to quickly skim through several publications before deciding to read one in full. It may therefore help to write out terms fully in abstract (For example, NMR, DPPH, ABTS).

3. Check the abbreviations throughout the manuscript and introduce the abbreviation when the full word appears the first time in the abstract and the remaining for the text and then use only the abbreviation (For example, reactive oxygen species (ROS), MTT, etc.,). Make a word abbreviated in the article that is repeated at least three times in the text, not all words to be abbreviated.

4. The time (season) of collection of this species should be added. As it is well-known that one of the factors affecting the constituents, is the time of collection, so please give top priority to this issue.

5. The authors may include the details of the quantity of fresh sample used and the quantity of the powder obtained (the ratio between fresh plant and powder) under the heading materials and methods.

6. When referring to SPSS versions beginning from 19, authors should cite ‘IBM SPSS Statistics for Windows, version 27 (IBM Corp., Armonk, N.Y., USA)'.

7. The references cited in the results may be shifted to any other part of the manuscript, since in the results, it should be the overall impact of the present findings and it should not support with others reference.

8. The conclusion provides a comprehensive overview of the study, but it could be more concise. Additionally, the conclusion could be strengthened by explicitly stating the implications of the research.

Reviewer #4: The title is insufficiently expressive of the content and needs to be rephrased

Background paragraphs or segments need to expand the content and more specific of the research work

Objective need to be clear, concise and expressive and embedded at the end of the intro

Materials and methods need to write the steps of each method and explain the abbreviations in detail.

Correction of equations used in calculations

The results in the tables (such as table 4) are incorrect and unintelligible and need to be reviewed with the typed text

Correct the correlation in table 6

Match the number of tables with the typed text

I recommended the author start the introduction with the main problem, such as colon cancer, and follow it with the solution (using OD SP from marine macroalgae).

The author usually refers to the cytotoxic as an anticancer effect, but we recommend using this expression only in case you tested the extract or compound against both cancer and a normal cell line.

I advise reducing the length of the objective paragraph of the current study.

How the authors identified and herbarium the collected algal species

I think 9 grams of seaweed is not enough for extraction, isolation and identification of active compounds in addition to obtaining enough crude extract to be tested in the bioassay.

We recommend rewriting the extraction process, especially the step of heating with dist. Water and the addition of ethanol (not clear)

How did the author confirm if the crude extract contained only SP or monosaccharides and other chemical compounds?

What is the method used for conversion of polysaccharide into monosaccharides before injection in HPLC?

The HPLC chromatogram and NMR charts not found

No tables for the monosaccharides analysed using HPLC (should provide the types, concentration, Rt, etc.)

No anticancer standard was used in this manuscript.

Reviewer #5: This manuscript explores the extraction, structural characterization, and bioactivity of sulfated polysaccharides from Gracilaria corticata with a focus on their antioxidant potential and pro-apoptotic effects in colon cancer cells. While the study demonstrates promising results, several methodological and interpretive aspects warrant deeper scrutiny. The following questions aim to probe the rigor, reproducibility, and translational relevance of the work.

1. How would the optimized extraction process perform at larger scales, and what steps are needed to ensure reproducibility across batches and environmental conditions?

2. Why does the monosaccharide composition differ so markedly from previous reports on G. corticata, and could extraction-induced degradation or selective enrichment explain this?

3. Beyond correlation, what experiments could directly prove that ROS elevation is indispensable for apoptosis induction in HT-29 cells?

4. The IC50 value is relatively high. Do the authors consider this a limitation for drug development, and how might bioavailability or delivery systems improve efficacy?

5. Which standard chemotherapeutics or targeted agents would be the most rational to test in synergy with SPs, and why?

6. Could chronic use of SPs as dietary supplements pose risks (e.g., over-suppression of ROS in normal physiology), and how might safety be assessed?

Reviewer #6: 1-What was the reason for choosing colon cancer cells for this study?

2- There are typographical and grammatical errors in the manuscript that need to be corrected.

3- The concentrations of the solutions used must be stated correctly. For example, the concentration of the MTT solution must be stated in the method section.

4- The significance level and number of replicates for Figure 4 should be provided.

5- A close examination of Figure 4 shows that the inhibitory effect of the extracted polysaccharides on ABTS free radicals is surprisingly similar to their inhibitory effect on DPPH free radicals at all concentrations. This result is unlikely due to the high similarity in percentages. Please review the results and graph again.

6- The discussion should be rewritten in a comparative manner with the results of previous studies.

Reviewer #7: 1- How the traces of ethanol was removed from extract. Please mention the amount of acetone in dried extract using GC-MS.

2- Figures and graphs. Figures and graphs should be set to a high resolution. Letters indicating statistical significance levels in Figure 4 should be added to the graphs.

3- Attention should be paid to spelling rules and punctuation in the text. Grammatical attention should be paid throughout the text.

4- HPLC analysis. The authors could provide more details about the analysis. Is there any reference standards? How the quantification was performed? Please mention the name of library used for identification of HPLC data.

5- While the "3.6.1. Cell culture and viability assay" section states that the SP treatment is 24 hours, it also states that it is 72 hours under the same heading. Which is correct? If the treatments were applied for 24 hours, why weren't the 48- and 72-hour treatments applied?

“The MTT assay was used to evaluate the viability of HT29 and CCD 841 cells following a 24-hour treatment with varying concentrations of SP. In 96-well microplates, cells were seeded in triplicate with a density of 5 × 10³ cells per well. Following an overnight incubation period, the cells were subjected to 72 hours of SP treatment at concentrations of 0, 12, 5, 25, 50, 100,200, and 200 μM”.

6- Supporting the gene expression studies with proteomic analysis can significantly contribute to demonstrating correlations and elucidating molecular mechanisms. Was a proteomic analysis such as a Western blot performed in this study?

7- Cytotoxicity assays should include a positive control group (e.g., for comparison with cells treated with chemotherapeutics such as cisplatin or paclitaxel). Was a positive control group used in this study?

8- Interesting references. https://doi.org/10.23751/pn.v19i1-S.5364,
https://doi.org/10.1016/j.fbio.2025.106082

6. PLOS authors have the option to publish the peer review history of their article (what does this mean?). If published, this will include your full peer review and any attached files.

Reviewer #1: No

Reviewer #2: No

Reviewer #3: **Yes:**Dr. A. Vijaya Anand

Reviewer #4: No

Reviewer #5: No

Reviewer #6: No

Reviewer #7: No

---

## [Author Response · Author response to Decision Letter 1]

23 Dec 2025

Hello Dear

With all the difficulties in my country to conduct scientific research, we tried to implement all the referees' comments. We hope to attract your attention.

Thank you so much...

---

## [Decision Letter · Decision Letter 1]

19 Jan 2026

PONE-D-25-45707R1ROS-mediated apoptosis in colon cancer cells induced by sulfated polysaccharides from Gracilaria corticataPLOS One

Dear Dr. roomiani,

Thank you for submitting your manuscript to PLOS ONE. After careful consideration, we feel that it has merit but does not fully meet PLOS ONE’s publication criteria as it currently stands. Therefore, we invite you to submit a revised version of the manuscript that addresses the points raised during the review process.

We look forward to receiving your revised manuscript.

Kind regards,

Prof. Fahrul Nurkolis

Academic Editor

PLOS One

Journal Requirements:

Additional Editor Comments:

Minor Revision!

Reviewers' comments:

Reviewer's Responses to Questions

**Comments to the Author**

1. If the authors have adequately addressed your comments raised in a previous round of review and you feel that this manuscript is now acceptable for publication, you may indicate that here to bypass the “Comments to the Author” section, enter your conflict of interest statement in the “Confidential to Editor” section, and submit your "Accept" recommendation.

Reviewer #1: All comments have been addressed

Reviewer #3: (No Response)

Reviewer #5: All comments have been addressed

Reviewer #7: All comments have been addressed

2. Is the manuscript technically sound, and do the data support the conclusions?

Reviewer #1: Yes

Reviewer #3: Partly

Reviewer #5: Yes

Reviewer #7: Yes

3. Has the statistical analysis been performed appropriately and rigorously? 

Reviewer #1: Yes

Reviewer #3: Yes

Reviewer #5: Yes

Reviewer #7: Yes

4. Have the authors made all data underlying the findings in their manuscript fully available?

Reviewer #1: Yes

Reviewer #3: Yes

Reviewer #5: Yes

Reviewer #7: Yes

5. Is the manuscript presented in an intelligible fashion and written in standard English?

Reviewer #1: Yes

Reviewer #3: Yes

Reviewer #5: Yes

Reviewer #7: Yes

6. Review Comments to the Author

Reviewer #1: Although the authors have addressed most of the previous comments and revised other sections accordingly, the quality of several figures (e.g., Figures 2, 3, and 5C) remains insufficient. The low resolution and unclear labeling hinder proper data interpretation. High-resolution figures with clearly legible annotations are still required.

Reviewer #3: 1. This suggestion is not carried out properly and it should be rectified. The use of abbreviations in the abstract may distract readers who wish to quickly skim through several publications before deciding to read one in full. It may therefore help to write out terms fully in abstract (For example, NMR, DPPH, ABTS, but given expansion in next para for DPPH and ABTS). The same may be considered in the title also.

2. This suggestion is not carried out properly and it should be rectified (Check the abbreviations throughout the manuscript and introduce the abbreviation when the full word appears first time in the abstract and the remaining for the text and then use only the abbreviation). For example, reactive oxygen species (ROS), MTT, and these types of corrections need to be checked all other abbreviations used in the manuscript.

3. This suggestion is not carried out properly and it should be rectified. When referring to SPSS versions beginning from 19, authors should cite ‘IBM SPSS Statistics for Windows, version 27 (IBM Corp., Armonk, N.Y., USA)'.

4. This suggestion is not carried out properly and it should be rectified. The references cited in the results may be shifted to any other part of the manuscript, since in the results, it should be the overall impact of the present findings and it should not support with others reference.

5. The conclusion seems in general and it should be cocise. All conclusions must be convincing statements on what was found to be novel impact based on the strong support of the data/results/discussion.

Reviewer #5: Thank you for the revision. The authors have adequately addressed the previous comments, and the manuscript is now clear and technically sound. I have no further substantive concerns and support acceptance.

Reviewer #7: I would like to note that the authors have made most of the previously indicated corrections, and I thank them for doing so.

7. PLOS authors have the option to publish the peer review history of their article (what does this mean?). If published, this will include your full peer review and any attached files.

Reviewer #1: No

Reviewer #3: **Yes:**Dr. A. Vijaya Anand

Reviewer #5: No

Reviewer #7: No

---

## [Author Response · Author response to Decision Letter 2]

13 Feb 2026

We would like to express our sincere gratitude to you and the reviewers for dedicating your valuable time to evaluating our manuscript and providing insightful and constructive comments. Your detailed feedback has been instrumental in guiding substantial improvements to the revised version.

The authors have carefully considered each comment and made every effort to address them thoroughly. We hope that the revised manuscript now meets your high standards and expectations.

Please note that the comments provided by Reviewer #2 were not implemented, as they appear to have mistakenly reviewed a different manuscript. Their remarks were unrelated to the content and scope of our submission.

Point-by-point responses to the reviewers’ comments are provided below. All revisions in the manuscript have been clearly highlighted.

With kind regards,

Dr. Laleh Roomiani

Reviewer #1: Although the authors have addressed most of the previous comments and revised other sections accordingly, the quality of several figures (e.g., Figures 2, 3, and 5C) remains insufficient. The low resolution and unclear labeling hinder proper data interpretation. High-resolution figures with clearly legible annotations are still required.

Response: Change was applied.

Reviewer #3: 1. This suggestion is not carried out properly and it should be rectified. The use of abbreviations in the abstract may distract readers who wish to quickly skim through several publications before deciding to read one in full. It may therefore help to write out terms fully in abstract (For example, NMR, DPPH, ABTS, but given expansion in next para for DPPH and ABTS). The same may be considered in the title also.

Response: Change was applied and marked in red.

2. This suggestion is not carried out properly and it should be rectified (Check the abbreviations throughout the manuscript and introduce the abbreviation when the full word appears first time in the abstract and the remaining for the text and then use only the abbreviation). For example, reactive oxygen species (ROS), MTT, and these types of corrections need to be checked all other abbreviations used in the manuscript.

Response: We sincerely thank the reviewer for highlighting this important issue regarding the consistency and correct introduction of abbreviations throughout the manuscript. We fully agree that a proper definition of abbreviations at their first occurrence, particularly in the abstract, is essential for clarity, readability, and adherence to scientific writing standards. Accordingly, the entire manuscript has been carefully re-examined to ensure that each abbreviation is introduced only once at its first appearance with the full term, after which the abbreviated form is used consistently.

3. This suggestion is not carried out properly and it should be rectified. When referring to SPSS versions beginning from 19, authors should cite ‘IBM SPSS Statistics for Windows, version 27 (IBM Corp., Armonk, N.Y., USA)'.

Response: Change was applied and marked in red.

4. This suggestion is not carried out properly and it should be rectified. The references cited in the results may be shifted to any other part of the manuscript, since in the results, it should be the overall impact of the present findings and it should not support with others reference.

Response: Change was applied and marked in red.

5. The conclusion seems in general and it should be cocise. All conclusions must be convincing statements on what was found to be novel impact based on the strong support of the data/results/discussion.

Response: Change was applied and marked in red.

---

## [Editor Report · Decision Letter 2]

17 Feb 2026

ROS-mediated apoptosis in colon cancer cells induced by sulfated polysaccharides from Gracilaria corticata

PONE-D-25-45707R2

Dear Dr. roomiani,

We’re pleased to inform you that your manuscript has been judged scientifically suitable for publication and will be formally accepted for publication once it meets all outstanding technical requirements.

Kind regards,

Fahrul Nurkolis

Academic Editor

PLOS One
---

## [Editor Report · Acceptance letter]

PONE-D-25-45707R2

PLOS One

Dear Dr. Roomiani,

I'm pleased to inform you that your manuscript has been deemed suitable for publication in PLOS One. Congratulations! Your manuscript is now being handed over to our production team.

Kind regards,

on behalf of

Dr. Fahrul Nurkolis

Academic Editor

PLOS One